

**Assessment of rockfall hazard on the steep-high slopes:**
**Ermenek (Karaman, Turkey)**
**Hidayet Taga and Kıvanç Zorlu**
Mersin University, Department of Geological Engineering, Mersin, Turkey
**Abstract**
Ermenek is one of the curious settlement areas because of its topographical features in
Karaman (Turkey). The city is located in northern side of the very steep cliffs formed by
jointed limestone which are suddenly increased from 1250 m to 1850 m. Moreover, these
cliffs having almost 90° slope dip are the main rockfall source areas due to their lithological
characteristics, climatic effects and engineering properties of rock units. Up to now,
depending on rockfall events, almost 500 residences were damaged severely, and losses of
lives were also recorded in Ermenek. The rockfall phonemon are initiated by discontinuities,
lithological changes, weathering and freeze-thaw process in the study area. In this study,
extensive fieldwork including determination of location and dimension of hanging, detached
and already fallen blocks, a detailed discontinuity survey, description of geological,
morphological and topographical characteristics was performed. Besides, rockfall hazard is
evaluated by two-dimensional rockfall analyses along 10 profiles. During the rockfall
analyses; run out distance, bounce height, kinetic energy and velocity of various size of
blocks for each profiles are determined by using RocFall v4.0 software. The results obtained
from rockfall analyses were used to map the areas possible rockfall hazard zones and
rockfall source areas were interpreted.
According to rockfall analysis, field study and laboratory testing, protective and preventive
recommendations can be suggested for the areas under rockfall threat. But, the most widely-
known remedial measures in literature such as trenches, retaining walls (barrier), wire
meshes, cable/streching nets and rock bolting etc. are not sufficient in the study area, due to
topographical, atmospheric and lithological features. For these reasons, firstly total
evacuation of the danger zone should be applied and then hanging blocks in the reachable
locations can be removed taking safety measures in this area to make it safer for the living
people.

Keywords: Hazard, Rockfall, Limestone, Zonation map, Ermenek.


## Introduction

Rockfall is a fast movement of the blocks which are detached from the bedrock along
discontinuities that slides, rolls or falls along vertically travels down slope by bouncing and
flying along trajectories (Varnes, 1978; Whalley 1984; Dorren 2003). Due to their high speed
and energy, rockfalls can be admissible as a substantially destructive mass movement
resulting in significant damage and loss of live. This movement is mainly controlled by the
geological conditions of the rock units, climatic influences and the weathering processes.
Besides, discontinuity patterns and the related intersections are also played an important role
of the size and shape of the detached blocks (Perret et al. 2004).
The slope characteristics are very significant factors for the rockfall events. The normal ($r_n$)
and tangential ($r_t$) components of coefficient of restitution, are related to the slope
characteristics that control behavior of the falling blocks and they are the most crucial input
parameters for rockfall analyses (Chau et al. 1996). Both components of coefficient of
restitution are related to material covering the surface, vegetation, surface roughness, and
radius of the falling rocks (Dorren et al., 2004). The coefficient of restitution with normal and
tangential components are best determined by the field tests and back analysis of the fallen
blocks. Although many researchers are revealed several techniques to determine the
coefficient of restitutions, these parameters should be identified individually for each side
because of the different geometrical features and mechanical properties of the slopes
(Agliardi and Crosta 2003; Dorren et. al, 2004; Evans and Hungr 1993; Robotham et al.
1995; Pfeiffer and Bowen 1989; Ulusay et al. 2006; Topal et al 2007; Topal et al., 2012, Buzzi
et al 2012).  On the other hand, slope inclination and slope properties are also affecting the
runout distances of the falling blocks (Okura et al., 2000). The slope surface of a hard rock
and free from vegetation cover is more dangerous then the surface covered by vegetation or
talus material because of the fact that it does not retard the movement of falling blocks.
To simulate fall of a blocks down a slope and to compute rockfall trajectories, various two
dimensional (2D) or tree dimensional (3D) and 2D-3D Discontinuous Deformation Analysis
(DDA)  programs  have been developed and tested during the last few years and many of
study considering with rockfall analyses and simulations are carried out. Additionally, the
rockfall susceptibility and hazard maps are produced using both two and tree dimensional
rockfall analysis technique considering with mostly traveling distance of falling blocks.
(Bassato et al. 1985; Falcetta 1985; Bozzolo and Pamini 1986; Hoek 1987; Pfeiffer and
Bowen 1989; Azzoni et al. 1995; Jones et al. 2000; Guzetti et al. 2002, Guzetti et al, 2003;
Agliardi and Crosta 2003; Schweigl et al 2003; Perret et al 2004; Yilmaz et al. 2008,



Tunusluoglu and Zorlu 2009, Binal and Ercanoglu 2010; Zorlu et. al 2011; Katz et al 2011;
Topal et al 2012;  Chen et al. 1994; Keskin 2013).

In this study, rockfall analyses are performed in Ermenek district located on very steep cliffs
considering past recorded phenomenon and recently ongoing threats of event (Fig 1). The
rockfalls occur very close to residential area and already damaged the houses and
unfortunately have been losses of lives. To reveal the rockfall potential of the study area, an
extensive field work including detailed discontinuity survey, determination of location and
dimensions of hanging, detached and already falling blocks, and also back analyses  was
carried out. Two dimensional rockfall analyses are conducted along 10 selected profile to
assess the block trajectories, runout distance, kinetic energy and bounce high of the blocks,
based on field and laboratory test data. Then a rockfall danger zonation map was produced
by means of the results obtained from rockfall analyses and areal extention of rockfall was
delineated. When considering location, climatic adversities and geological factors of the study
area, some remedial measures can be arguable. Despite the unfavourable conditions,
possible remedial measures are suggested for the study area.

**Geological Settings**

The Ermenek basin is one of the Neogene intramontane molasse basin formed in Central
Taurides, the orogenic belt's segment streching between the Isparta angle to west and the
Ecemi  Fault to the east (Özgül, 1976; Ilgar and Nemec, 2005). The Ermenek basin and the
adjacent Mut Basin lies between the Cukurova basin complex to the east and the Antalya
basin complex to the west and is situated within the central part of the Taurides, an E-W
trending orogenic belt that originated through compressive deformation during the initial
stage of closure of the southern branch of the Neo-Tethyan ocean in the Early Cenozoic
(Safak et al. 1997). The basins evolved as extensional grabens related to preexisting
fractures. Depozition resumed in Early Miocene time, with Mut basin hosting alluvial
sedimentation and the Ermenek basin becoming a large clastic lake. The two basins, formed
as separate interamontane depressions, were then inundated by the sea near the end of the
Early Miocene and jointly covered with an extensive, thick succession of late Burdigalian-
Serravalian carbonates, including reefal and platform limestones (Ilgar and Nemec, 2005).

The tectonic history of Southern Turkey can be summarised into three major periods; (1) Late
Palaeozoic to Middle Eocene: formation of the Tethyan orogenic collage. (2) Middle Eocene
to Middle Miocene: Tauride Orogeny during continued north-south convergence and collision;
migration of deformation front south of Turkey. (3) Late Miocene to recent: collision of




Eurasia with the Arabic Plate and start of the Neotectonic Regime (Bassnt et al 2005). Due to
this complex tectonic movement the Taurus Belt exhibits very complicate stratigraphic
sequence and litological diversity (Fig 2).

The basement of the Ermenek basin consists of Paleozoic and Mesozoic units, which are
generally exposed at the southern part of the basin. Palaeozoic units compise of shale,
limestone, dolomitic limestone, and quartzite. While Lower-Middle Triassic units contains
limestone, shale; Upper Triassic units consists of sandstone, conglomerate and limestone;
Jura-Cretaceous time is represented by dolomitic limestone (Gul and Eren, 2003). Eocene
and Palaeocene sedimentary units contain fossiliferous limestone (Tepebasi Formation)
unconformably overlie the Cretaceous limestone and ophilotic melange. Oligocene lacustrine
deposits represent by Pamuklu Formation including coal layer as Yenimalle Formation,
overlies unconformably Eocene-Oligocene units in the area. The Yenimahalle Formation has
a great lateral extentention in the Ermenek basin consists of six main facies association,
which range from alluvial to offshore lacustrine deposits, up to 300 m in thickness. Middle
and Upper Miocene units unconformably overlie the Lower Miocene unit in the basin are
characterized by Mut, Köselerli and Tekecati Formations. Koselerli Formation comprises
claystone, limestone, clayey limestone, gravelly sandstone and marl deposits representing
centre of the reef (reef core facies). Mut formation also consist of reef limestones deposits in
shallow marine environment including limestone with clayey or fossiliferous, and distinctive
patch reefs are common in this formation (Gul and Eren, 2003). The last unit of the Miocene
age sequence of the basin is Tekecati Formation consists of limestone, fossiliferous
limestone, clayey limestone and mudstone as assessed typically shallow sea sediment
belong to a reefal environment (Yurtsever et al. 2005). All these formations of Middle and
Upper Miocene also interfinger and they have transitional contacts with each other (Fig 3).

A Digital Elevation Model (DEM) of the study area was constructed by implementation of
contour lines of 1:25,000 scale topographic maps with an equidistance of 10 m.  When
considering DEM, the altitude values of the northern and the south-eastern parts of the study
area vary from 1,200 to 1,860 m (Fig 4a), slope gradients exceed 90º from 0º (Fig 4b) and
the general physiographic trend of the study area is about S-SE (Fig 4c)

**Field investigation and engineering properties of the rock**

Rockfall events are observed in the very steep cliffs formed by jointed limestone which are
suddenly increased from 1250 m to 1850 m. The limestone of Mut Formation is not form from
a single lithological property it is also formed by succession of different lithologies which is





one of the triggering factor of the rock fall events. Owing to its complex lithological structure,
the field studies are also carried out more detailed considering with lithological differences. A
systematic sampling was conducted to determine the lithological and geomechanical
properties of Mut formation with different facieses. Petrographic investigations of the
limestone specimens from the systematic sampling along X-X' line (Fig 5) of formation
consists of routine observations under polarized microscope. According to the results of the
petrographic analyses, the specimens are formed by four lithological units such as,
fossiliferous limestone, claystone-marl, clayey limestone and limestone. The results of
petrographic thin-section analyses are summarized in Table 1.

The X-ray diffraction analyses (XRD) are also applied to the specimens to assess the relative
quantity of minerals (Table 1). The XRD diffractograms are obtained at General Directorate of
Mineral Research and Exploration X-Ray Laboratory. The X-ray diffraction and the thin-
section analyses results show it is obvious that Mut formation arise four different litological
units.

During the field studies a series of systematic scan-line surveys were carried out to
determine the orientation and spacing of discontinuities based on ISRM (1978) and ISRM
(1981). According to scan-line survey, five main discontinuity sets were determined via
contour diagrams using a computer program, name of DIPS 5.1 (RocScience Inc. 2006). The
dip and dip direction of values of the major sets are 86/154, 85/210, 87/173, 84/077 and
55/155 (Fig 6). The discontinuities have high persistence ( 20 m), very tight to very open
aperture (from 0.1 mm to  10 cm) without infilling. The discontinuity surfaces are rough,
undulating and groundwater seepage is not existed through discontinuities surface.  The
average spacing value of discontinuities is determined as 170 cm. and the discontinuity
spacing histogram is given in Fig 7.

Kinematic analyses of the discontinuities are conducted for western, northern and eastern
slopes of the study area. Kinematic analyses show that two different failure types observed
on the slopes. Although sliding is encountered as a main failure type on the each slope,
toppling type of failure is occurred only western and northern part (Fig 8).
During the field work, already fallen and hanging blocks in various dimensions were observed
in the study area. For real approaches at rockfall modeling, size, location and runout distance
of fallen blocks were determined (Fig 9). In addition to various sizes of hanging blocks,
different rockfall source area was also observed during the field study (Fig 10). Besides,
block samples were taken in the field for laboratory test. While taken block sample,
systematic sampling was carried out from bottom to top of slopes due to its different



lithological and mineralogical features of Mut formation. The tests performed in the laboratory
are unit weight, apparent porosity, void ratio, water absorption by weight, water absorption by
volume and uniaxal compression strength for each sampling zone.  When applying the tests,
the procedures suggested by ISRM (1981) suggested methods are taken into consideration.
The average unit weight of limestone samples (23.9 kN/m$^3$) were the highest than the
fossiliferous limestone (22.2 kN/m$^3$), claystone-marl (20.4 kN/m$^3$), and clayey limestone (21.5
kN/m$^3$) sample. The uniaxial compressive stress values of samples were found vary in a
large range. The average uniaxial compressive strength values of limestone, fossiliferous
limestone and clayey limestone were 55 MPa, 48 MPa and 36 MPa respectively. The
standard core sample can not be extracted from highly weathered zones of claystone-marl
for uniaxial compression tests. To cope with this difficulty, the Schmidt hammer index test
was performed in the field. The average Schmidt hammer rebound number of the claystone-
marl was obtained as 33 and the uniaxial compression strength value was found as 22 MPa
indirectly. The results obtained from the tests with statistical evaluations are given in Table 2.

**Rockfall analyses**

Various two or three-dimensional computer program are existed to simulate fall of boulder
and compute rockfall trajectories, (Bassato et al. 1985; Falcetta 1985; Bozzolo and Pamini
1986; Hoek 1987; Pfeiffer and Bowen 1989; Azzoni and de Freitas 1995; Jones et al. 2000;
Guzzetti et al. 2002). In this study, rockfall simulations of Ermenek steep cliffs carried out
using Rockfall V.4 software (Rocscience Inc. 2002). Rockfall V.4 is a two-dimensional
software program performing statistical analyses of rockfall and calculation engine behaves
as if the mass of each rock is concentrated in an extremely small circle. While simulate
rockfall trajectories, any size or shape effects must be accounted for by an approximation of,
or adjustments to, other properties (Rockscience Inc. 2002). Some crucial parameters are
required to design block trajectories and rockfall analysis, the coefficient of restitution (normal
and tangential), slope geometry, roughness of slope and weight of hanging blocks. The slope
geometry is revealed from 1/1.000 scale topographic map. When considering lithological
features, distance from settlement district and location of rockfall source areas, ten slope
profile selected for rockfall simulation analysis (Fig 11).  In the field study, hanging blocks are
determined and weight of reachable block is calculated by using unit weight and volume of
the rock (Fig 12). The hanging or detached blocks had various dimensions due to the
discontinuity orientation, spacing and their mineralogical composition affected by weathering
processes. The calculated hanging blocks weights vary between 75 kg and 9.800 kg for
different rockfall source areas (see Fig 10). For selected ten profiles, different rock masses
(100 kg, 1.000 kg and 10.000 kg) were used in the rockfall analyses considering block sizes


which are ideally represented field conditions. Initial velocity of blocks was preferred 0 m/s in
the analyses considering the situation of the each block.

The slope characteristics are very important factors for the rockfall event because of the fact
that the slope properties control the behavior of the falling blocks as runout distance of the
blocks (Okura et al. 2000). The slope surfaces was played a considerable role in movement
of falling or rolling blocks moving through the slopes. The slope faces are free from
vegetation cover, they do not retard the movement of blocks. In this case, the blocks can be
reached farther distance, on the contrary of the surface covered by vegetation or talus
material. Because, the vegetation or talus material absorbs a high amount of the energy of
the falling rock and will probably stop it (Hoek 2007). The retarding capacity of the slope
surface material is expressed mathematically normal ($R_n$) and tangential ($R_t$) coefficient of
restitution are affected by the composition of the material covering the surface and slope
roughness. The coefficient of restitutions can be obtained from back analyses in the field or
theoretical estimations (Agliardi and Crosta 2003; Dorren et. al, 2004; Evans and Hungr
1993; Robotham et al. 1995; Pfeiffer and Bowen 1989; Ulusay et al. 2006; Topal et al. 2007).
Back analyses were performed to determine the coefficient of restitution with ten blocks in the
field considering the size and the shape of the blocks and the slope characteristics (Fig 13).
The results of the analyses, normal and tangential coefficients of restitution values belong the
fallen rocks are determined as (0.33±0.04) and 0.63±0.19) respectively. In addition to
coefficients of restitution, friction angle was determined by field back analyses as 32.5°.
During the rockfall analyses 1.000 rock blocks were thrown. The slope roughness which is
another input parameter of rockfall simulation analyses was taken as 2° in based on the
angle between rough surfaces. The input parameters used for rockfall analyses are given in
Table 3.

Rockfall simulation analyses were performed ten profiles as mentioned above. The limestone
and fossiliferous limestone units resisting against weathering, upper zones of weaker
lithological unit claystone-marl accepted as rockfall source areas, based on field conditions
(Fig 14). During the rockfall analyses, different rock masses (100 kg, 1.000 kg, 10.000 kg)
were used for each profiles considering the real masses of hanging blocks in the study area.
One of the typical examples of a rockfall trajectory is given in Fig. 15 .The runout distance,
bounce height, kinetic energy and velocity of the blocks were predicted by rockfall analyses.
According to the results of the analyses, maximum runout distance reaches 660 m, kinetic
energy 1.750.000 kJ and velocity is 46.3 m/s for the free falling of the 1000 kg blocks. The
results of analyses are summarized in Table 4. A rockfall danger zone map was produced by
using the results obtained from rockfall analyses considering maximum runout distance of





falling blocks Fig 16. According to map, areal extention of all blocks for each profile would be
able to reach to the roads or settlement area. It is apparent that the settlement area was
located in the danger zone. Although some preventative measures can be applied to reduce
rockfall hazard, it was directly depend on topographical and lithological factors of the
potential rockfall source area. Also the aesthetic and socio-economic conditions were limited
to the existing preventative measurements. Construction of trenches, retaining walls (barrier),
wire meshes, cable/streching nets, rock bolting and evacuation of the danger zone can be
used as preventive measures in the rockfall areas. But, the most widely-known remedial
measures in literature are not proper in the study area, due to topographical and lithological
features. Thus, to apply trenching and fencing is not possible due to the big size of hanging
blocks have relatively high kinetic energy and bounce height. Rock bolting can not be applied
higher elevations because the slopes have considerably steep cliffs and large block sizes.
Therefore, it is recommended that the hanging blocks in the reachable locations should
removed taking safety measures. Although total evacuation of the danger zone is not
preferred by the residents, in opinion of the authors of this study, it is indispensible in the
study area.

**Results and conclusions**

Ermenek is a spectacular settlement area located in very steep cliffs with a height of 1850 m.
The settlement was subjected to rockfall event several times and resulted in loss of life and
property. During the fieldwork and depending on laboratory test results, the rockfalls were
initiated by discontinuities, weathering process and characteristics of limestone having
different lithological facieses. Considering the scan-line survey, five main discontinuity sets
were determined. To understand of rockfall mechanism, relevant with lithological features, X-
ray diffraction and thin-section analyses were taken into consideration and revealed that the
limestone formation are formed by four lithological units such as, fossiliferous limestone,
claystone-marl, clayey limestone and limestone. Thus rockfall occurs uppermost level of
limestone and fossiliferous limestone due to existence of weaker claystone-marl at the lower
level of the facies.

Two dimensional rockfall analyses were performed using the data collected from field study
and laboratory test results along 10 profiles. Rockfall analyses were indicated that the roads
and the settlement area were remaining in the rockfall danger zone. Considering
topographical and lithological limitations, commonly used remedial measures are not

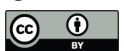


preferred in present study.  Total evacuation and cleaning the loose blocks in accessible
locations are recommended.

**Acknowledgment**

This research is supported by The Scientific and Technological Research of Turkey,
TUBITAK (Project No:107Y071).

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



List of Tables
































**Table 1** Results of the thin-section petrographic and X-ray analyses

| Specimen No | Petrographic description | X-ray analsis result | Microscopic photograph |
|---|---|---|---|
| **HK05-1** | Limestone | Calcite, Quartz | |
| **HK05-2** | Clayey Limestone | Calcite, Qartz Chlorite, Dolomite | |
| **HK05-3** | Claystone-Marl | Calcite, Dolomite, Simectite | |
| **HK05-5** | Fossiliferous Limestone | Calcite, Dolomite | |






**Table 2** Laboratory and field test results

| | Limestone | | | | Clayey limestone | | | | Claystone-Marl | | | | Fossilliferous Limestone | | | |
|---|---|---|---|---|---|---|---|---|---|---|---|---|---|---|---|---|
| | Max. | Min. | Average | Standart Deviation | Max. | Min. | Average | Standart Deviation | Max. | Min. | Average | Standart Deviation | Max. | Min. | Average | Standart Deviation |
| Unit weight (kN/m³) | 24,3 | 23,2 | 23,9 | 0,31 | 22,0 | 21,2 | 21,5 | 0,22 | 20,7 | 20,2 | 20,4 | 0,19 | 23,6 | 20,5 | 22,2 | 0,99 |
| Void ratio (%) | 7,26 | 3,70 | 5,33 | 1,16 | 19,05 | 16,62 | 17,67 | 0,77 | 24,37 | 20,02 | 21,47 | 1,28 | 21,99 | 7,36 | 11,88 | 3,40 |
| Porosity (%) | 6,77 | 3,57 | 5,05 | 0,96 | 16,00 | 14,25 | 14,98 | 0,55 | 19,59 | 16,68 | 17,67 | 0,86 | 18,02 | 6,85 | 10,47 | 3,77 |
| Water absorption by weight (%) | 2,86 | 1,45 | 2,07 | 0,42 | 7,39 | 6,48 | 6,81 | 0,29 | 9,44 | 7,89 | 8,48 | 0,44 | 8,63 | 2,86 | 4,70 | 1,97 |
| Water absorption by volume (%) | 6,77 | 3,57 | 5,05 | 0,96 | 16,00 | 14,25 | 14,98 | 0,55 | 19,59 | 16,68 | 17,67 | 0,86 | 18,02 | 2,86 | 10,47 | 3,77 |
| Uniaxial compressive strength (MPa) | 73,4 | 46,2 | 55,3 | 10,58 | 39,8 | 32,4 | 36,1 | 2,57 | 27,4 | 18,1 | 22,2 | 3,90 | 59,6 | 29,4 | 48,1 | 12,08 |
| Schmidt hammer rebound number | | | 55 | | | | 39 | | | | 25 | | | | 48 | |
| Number of Samples | | | 13 | | | | 11 | | | | 11 | | | | 10 | |






**Table 3** Input parameters used in the rockfall analyses

| Parameter | Value |
|---|---|
| Number of rockfall | 1000 |
| Minumun velocity cut off (m/s) | 0.1 |
| Coefficient of normal restitution | 0.33±0.04 |
| Coefficient of tangential restitution | 0.63±0.19 |
| Friction angle( ) | 32.5˚ |
| Slope roughness | 2 |
| Initial velocity (m/s) | 0 ± 0.5 |




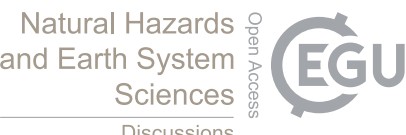
**Table 4** Results of the rockfall analyses

| Profile Number | Maximum Slope Height (m) | Weight of block (kg) | Runout distance (m) | | Bounce height (m) | | Kinetic energy(kJ) | | Velocity (m/s) | |
|---|---|---|---|---|---|---|---|---|---|---|
| | | | Max | Min | Max | Min | Max | Min | Max | Min |
| **1** | 88 | 100 | 683 | 170 | 7 | 0,5 | 14000 | 1000 | 16 | 1 |
| | | 1000 | 585 | 195 | 3 | 0,5 | 120000 | 5000 | 14 | 2 |
| | | 10000 | 480 | 160 | 7,25 | 0,5 | 1400000 | 10000 | 16 | 2 |
| **2** | 33 | 100 | 535 | 233 | 2,8 | 0,5 | 2200 | 1025 | 6,3 | 2 |
| | | 1000 | 280 | 233 | 3 | 0,5 | 18200 | 1247 | 7,21 | 1,02 |
| | | 10000 | 277 | 222 | 3,2 | 0,45 | 62033 | 6300 | 3,27 | 1,03 |
| **3** | 334 | 100 | 272 | 115 | 13,25 | 1,23 | 14027 | 60150 | 16,29 | 8,23 |
| | | 1000 | 275 | 48 | 19 | 2 | 1750000 | 7350 | 46,3 | 8,23 |
| | | 10000 | 263 | 72 | 108,5 | 13,5 | 11200000 | 860000 | 63,5 | 9.87 |
| **4** | 145 | 100 | 323 | 75 | 68,3 | 4,3 | 34500 | 6350 | 28,43 | 3,45 |
| | | 1000 | 312 | 80 | 11,8 | 3,05 | 583400 | 54000 | 34,42 | 3,02 |
| | | 10000 | 325 | 82 | 13,8 | 6,32 | 6973000 | 425000 | 33,25 | 2,1 |
| **5** | 123 | 100 | 273 | 32 | 57,32 | 11,3 | 64300 | 8920 | 36,32 | 12,32 |
| | | 1000 | 281 | 40 | 68,32 | 4.06 | 670000 | 33000 | 33,24 | 4,2 |
| | | 10000 | 283 | 45 | 68,3 | 4,23 | 6270000 | 4350000 | 33,5 | 6,7 |
| **6** | 103 | 100 | 235 | 12 | 8 | 2 | 102500 | 5000 | 46 | 3,8 |
| | | 1000 | 232 | 11 | 6,4 | 1,8 | 958000 | 5800 | 45 | 4,2 |
| | | 10000 | 88 | 43 | 18 | 3,2 | 10500000 | 560000 | 43,5 | 3,8 |
| **7** | 336 | 100 | 7,8 | 2,8 | 1,2 | 0,2 | 8320 | 1823 | 12 | 5 |
| | | 1000 | 7,5 | 1 | 1,1 | 0,18 | 83000 | 31000 | 11,5 | 7 |
| | | 10000 | 7,6 | 1,2 | 0,8 | 0,2 | 7000000 | 480000 | 11,5 | 3 |
| **8** | 104 | 100 | 77 | 23 | 15 | 3,8 | 14300 | 3200 | 16,7 | 2,9 |
| | | 1000 | 76 | 24 | 55 | 12 | 870000 | 38000 | 45 | 5 |
| | | 10000 | 34 | 27 | 57 | 8 | 8650000 | 43000 | 42 | 7 |
| **9** | 95 | 100 | 249 | 215 | 18 | 3 | 72000 | 11000 | 38 | 11,5 |
| | | 1000 | 248 | 235 | 21 | 3,2 | 630000 | 42000 | 36 | 8 |
| | | 10000 | 248 | 234 | 24,5 | 0,5 | 7120000 | 1050000 | 37 | 12 |
| **10** | 68 | 100 | 670 | 480 | 1,9 | 0,5 | 24500 | 1800 | | |
| | | 1000 | 660 | 510 | 2,4 | 0,5 | 653000 | 43500 | 34,5 | 5,2 |
| | | 10000 | 660 | 512 | 3 | 0,25 | 6250000 | 254000 | 34 | 4,3 |



**List of figures**

**Fig.1** Location map of the study area
**Fig.2** General view of the study area
**Fig.3** Geological map of the Ermenek region
**Fig.4** Maps of the study area, and their distributions of (a) altitude, (b) slope gradient,
(c) slope aspect
**Fig.5** Systematic sampling locations along X-X' line
**Fig.6** Contour diagram of major discontinuity sets
**Fig.7** Discontinuity spacing histogram
**Fig.8** Kinematic analysis results of the slopes
**Fig.9** Fallen blocks and their sizes
**Fig.10** Rockfall source areas of the Ermenek region
**Fig.11** Rockfall profiles analyzed in the study area
**Fig.12** Photograph showing hanging blocks and their location in the study area
**Fig.13** Back analyses in the field to determine the coefficient of restitutions
**Fig.14** Distribution of fallen and hanging blocks of the rockfall source areas
**Fig.15** An example for the rockfall analyses results (a) runout distance (b) total
kinetic energy (c) bounce height (d) the typical rockfall trajectory
**Fig.16** The map showing the rockfall danger zone of study area

















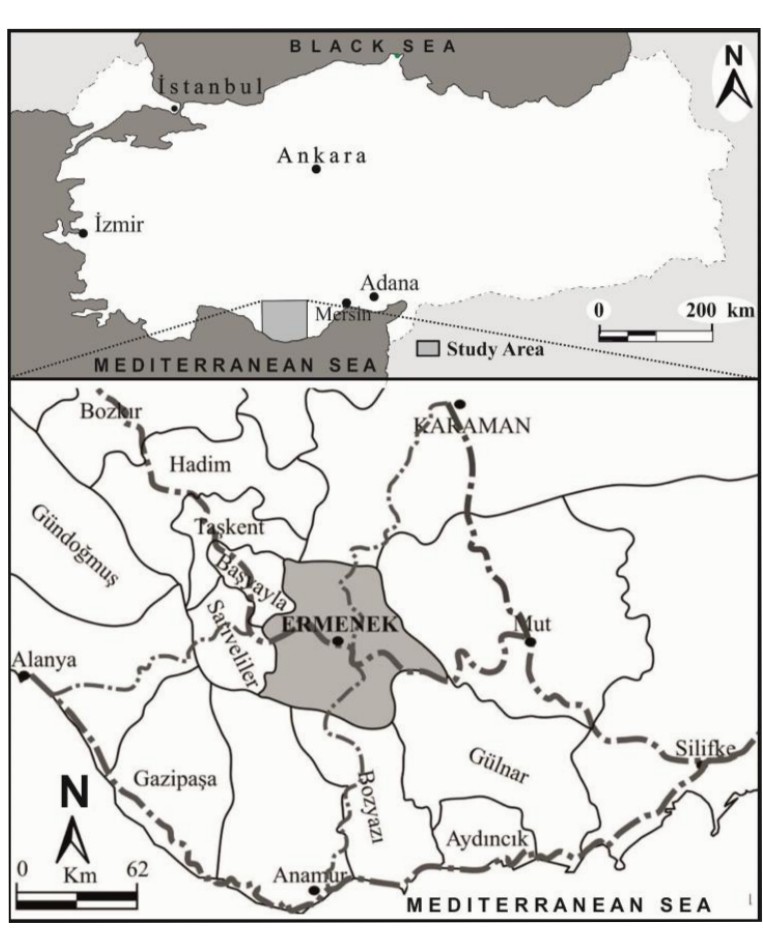


495                            **Fig.1** Location map of the study area






**Fig.2** General view of the study area





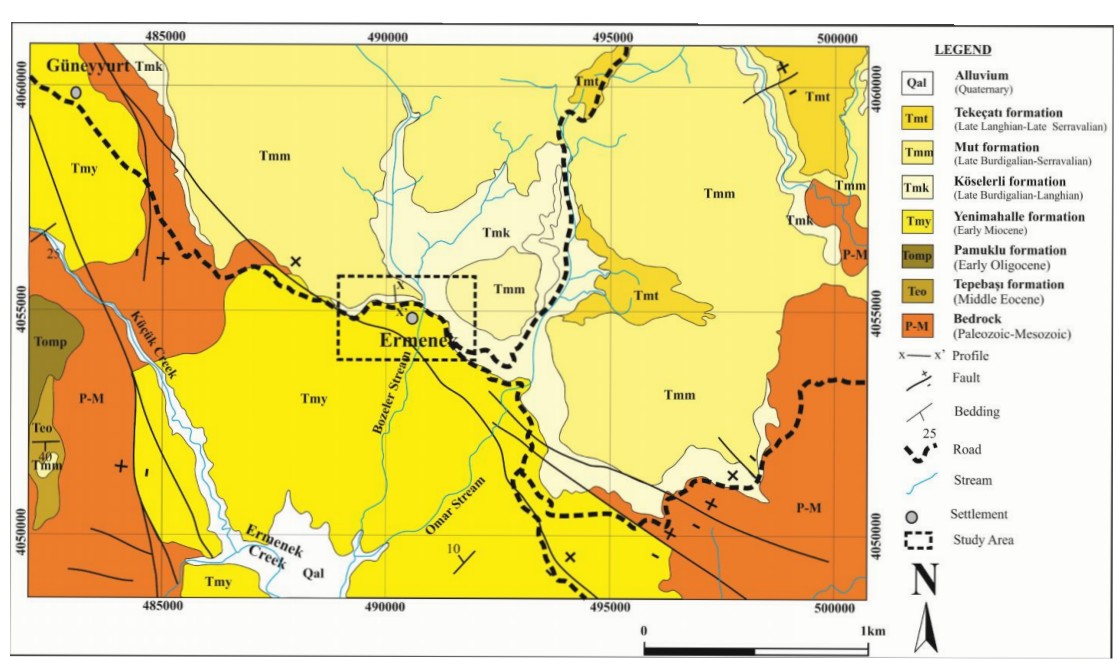

**Fig.3** Geological map of the Ermenek region




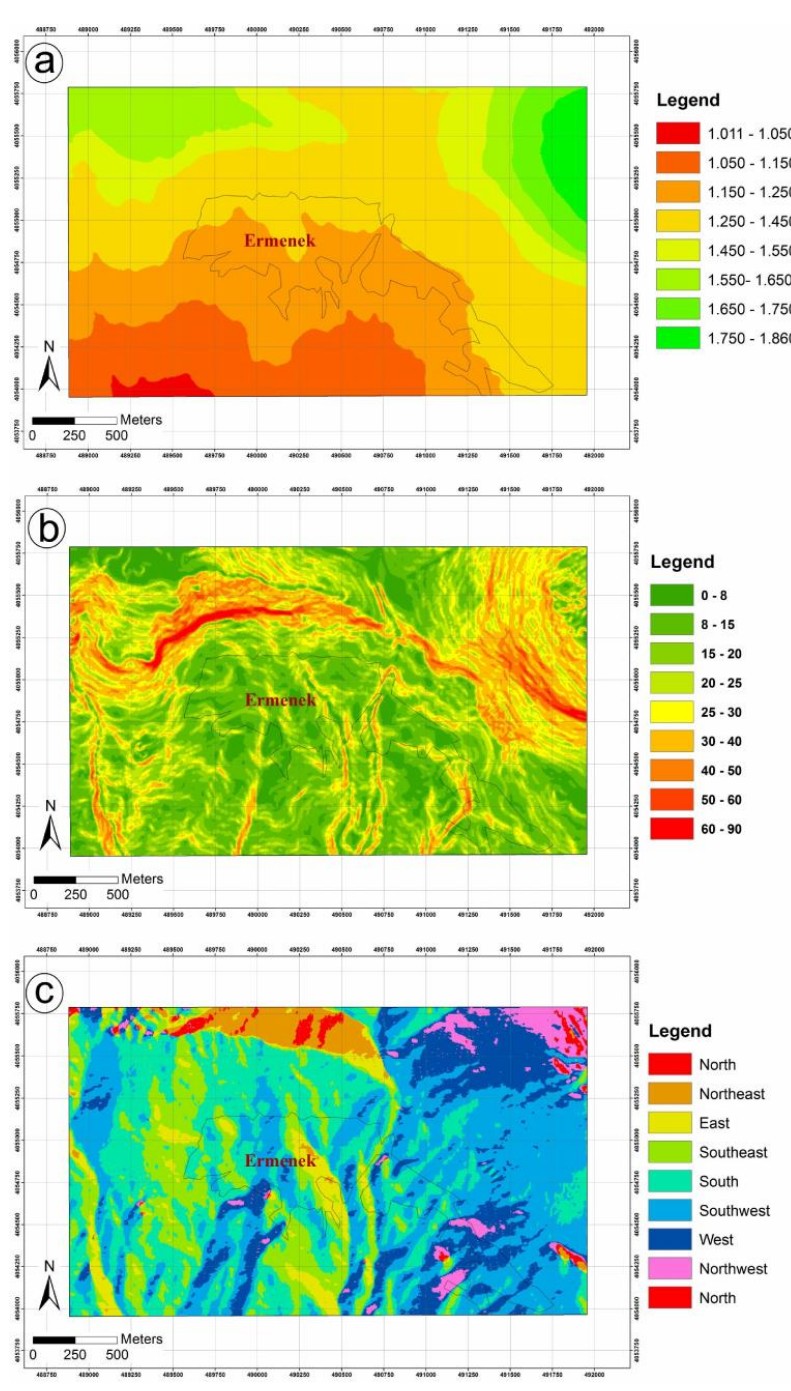



**Fig.4** Maps of the study area, and their distributions of (a) altitude, (b) slope gradient,
(c) slope aspect




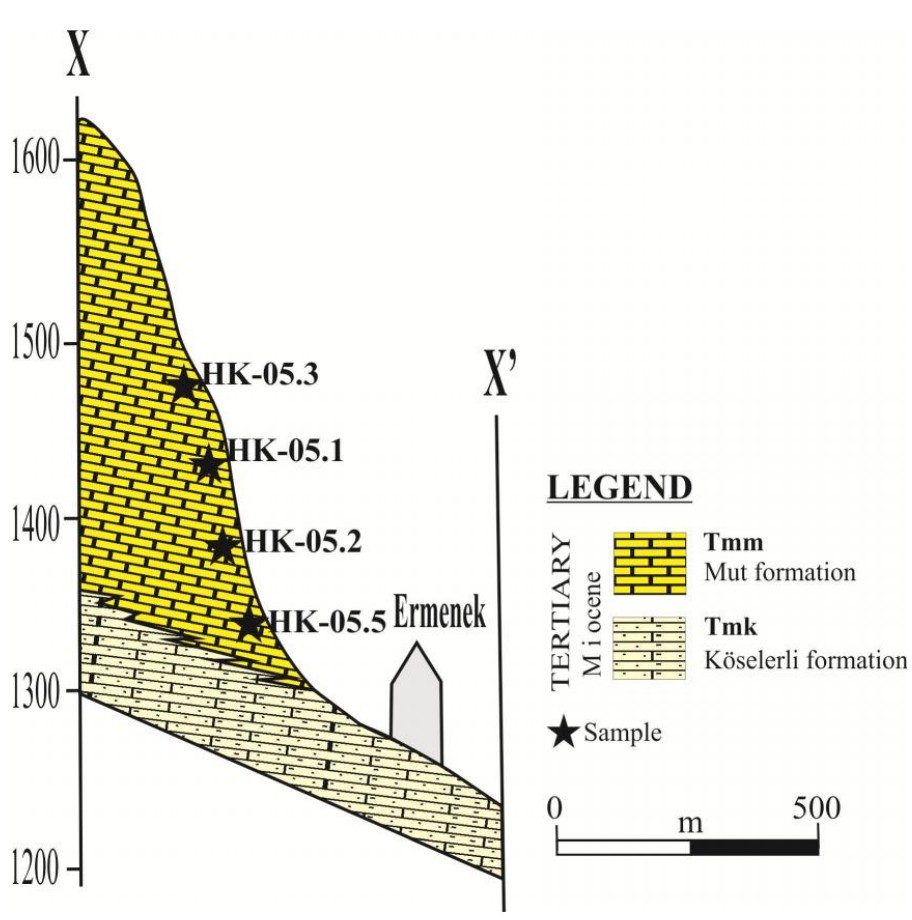

**Fig.5** Systematic sampling locations along X-X' line




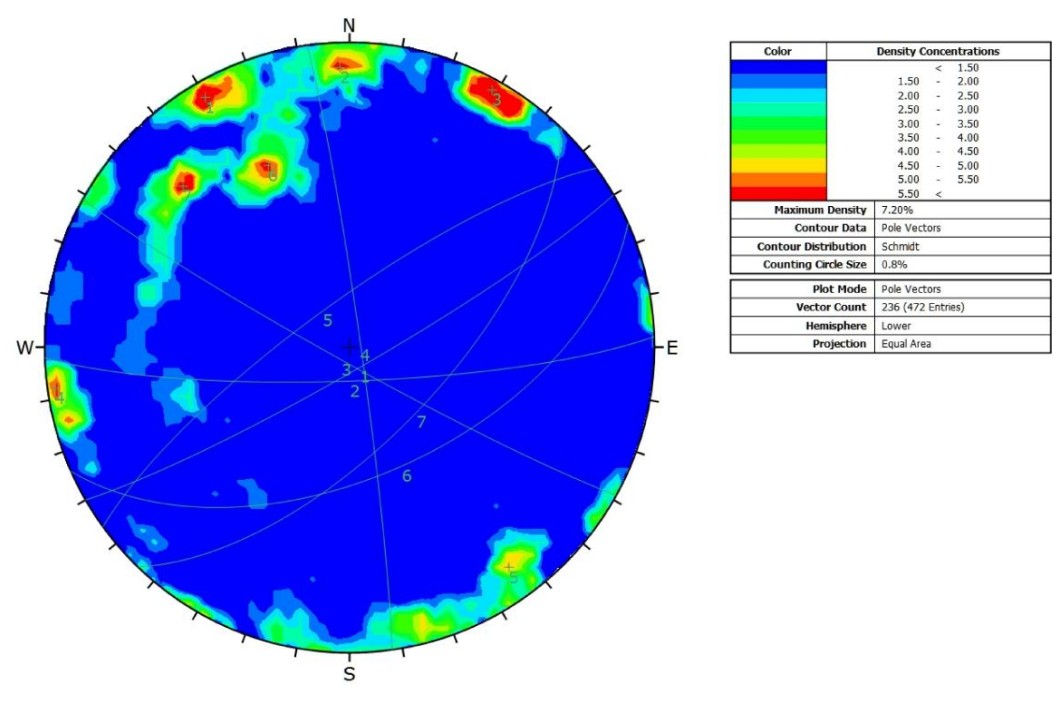



**Fig.6** Contour diagram of major discontinuity sets



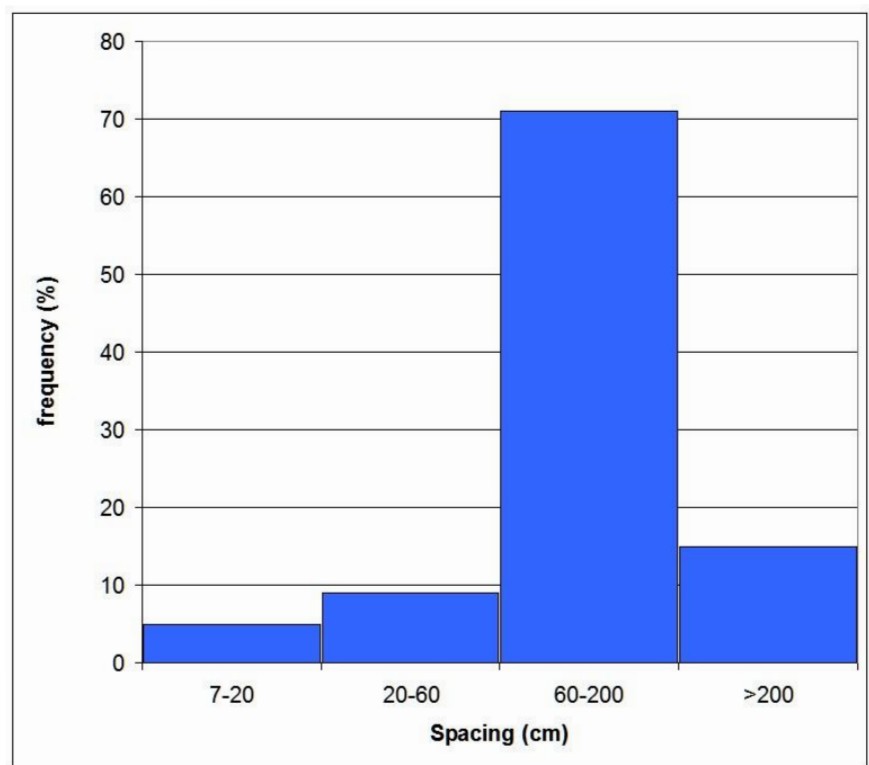



519         **Fig.7** Discontinuity spacing histogram





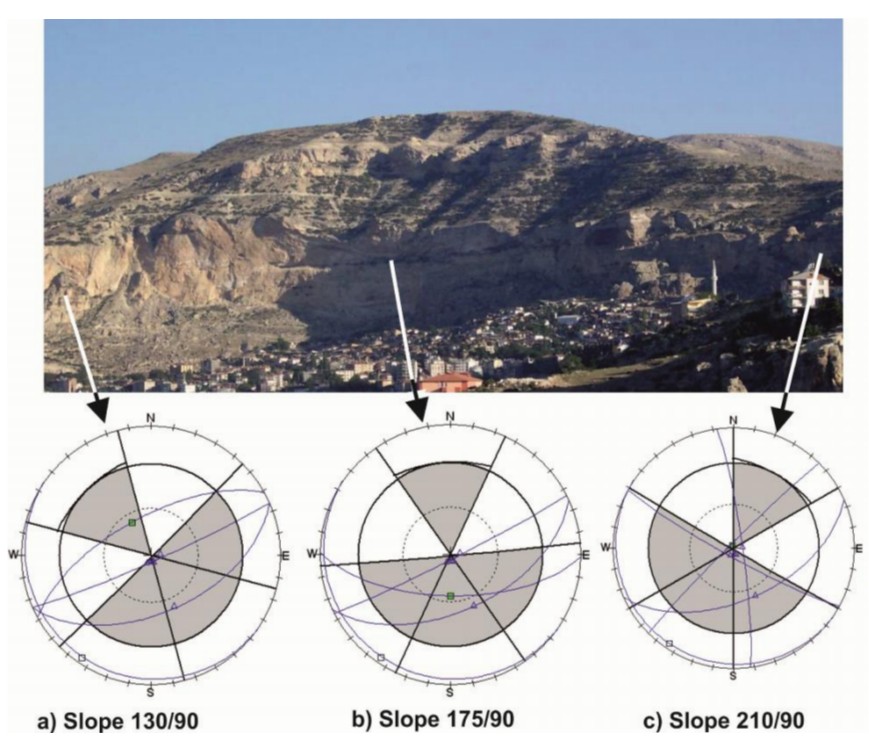

a) Slope 130/90      b) Slope 175/90      c) Slope 210/90



**Fig.8** Kinematic analysis results of the slopes







**Fig.9** Fallen blocks and their sizes






**Fig.10** Rockfall source areas of the Ermenek region






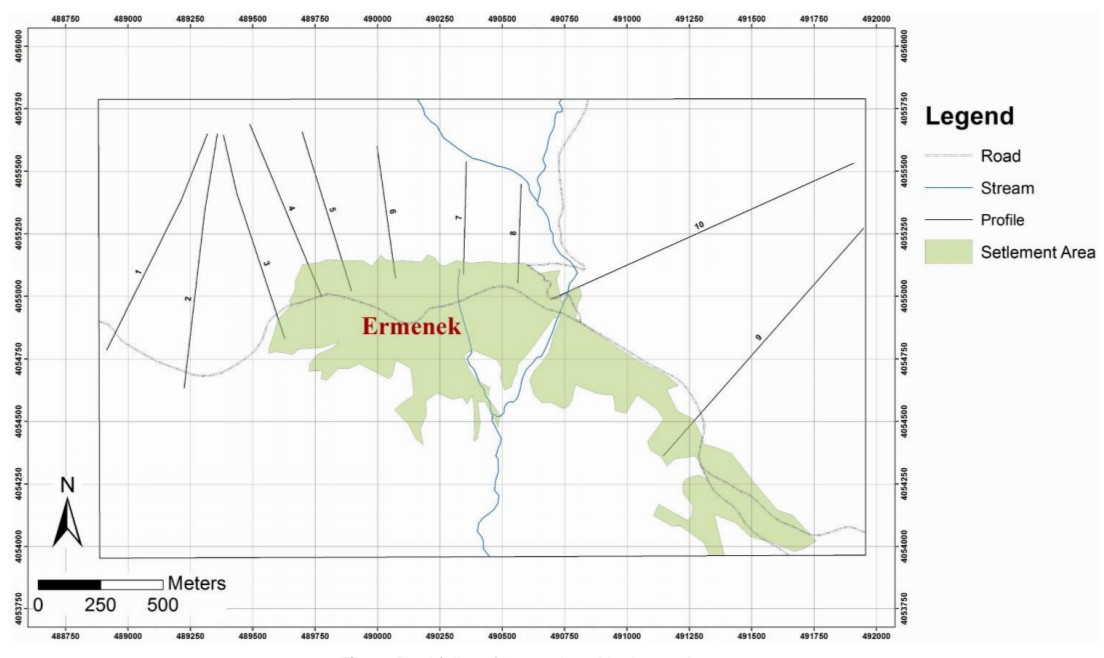

**Fig.11** Rockfall profiles analyzed in the study area




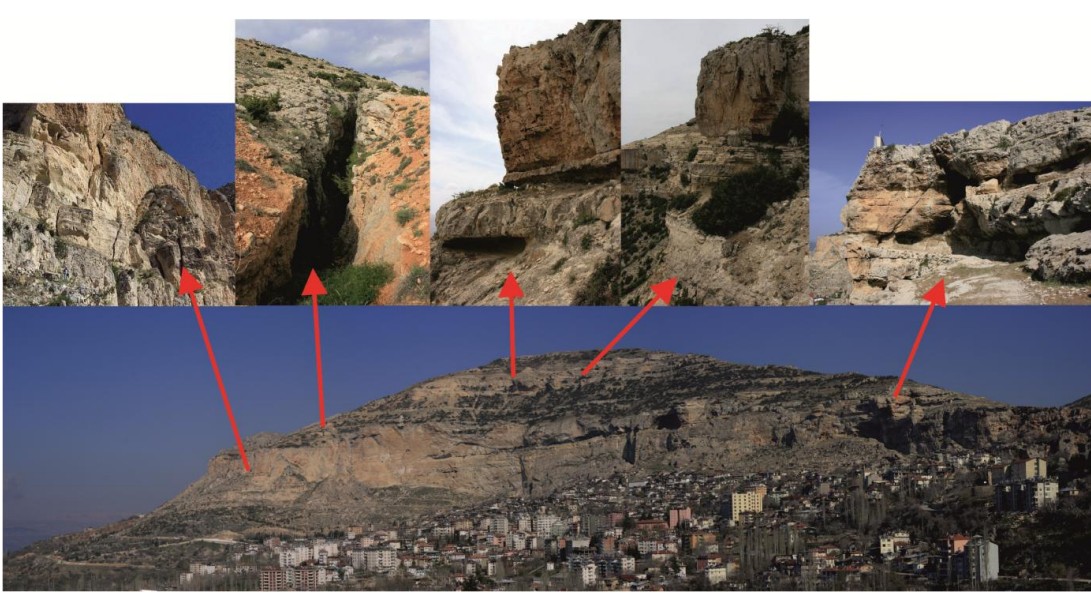

538            **Fig.12** Photograph showing hanging blocks and their location in the study area






**Fig.13** Back analyses in the field to determine the coefficient of restitutions




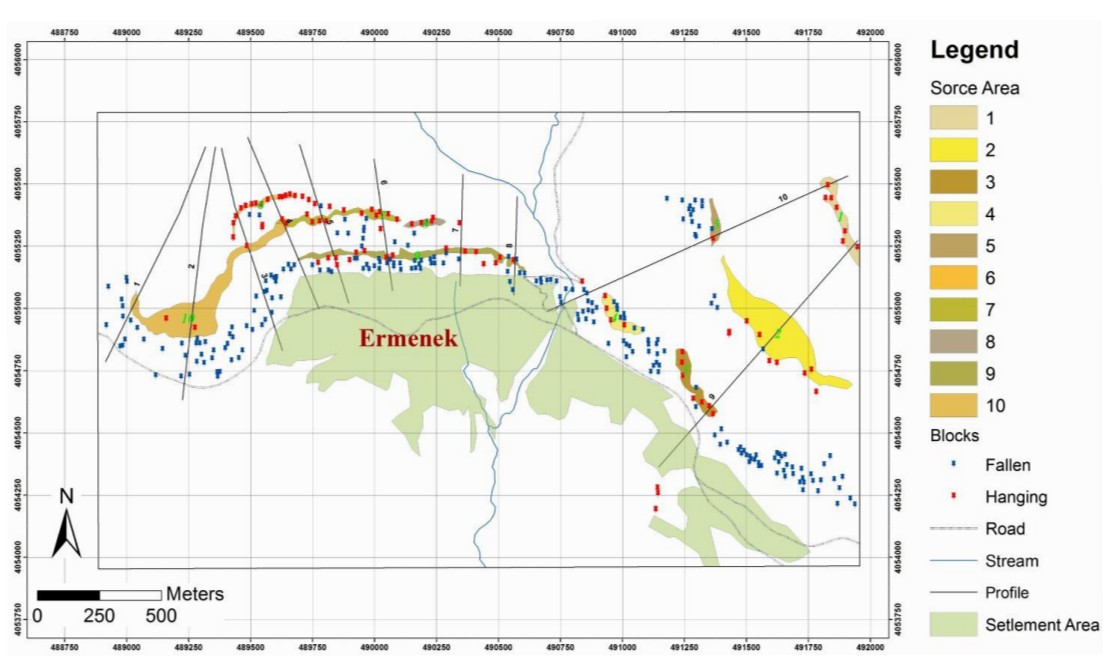

**Fig.14** Distribution of fallen and hanging blocks of the rockfall source areas




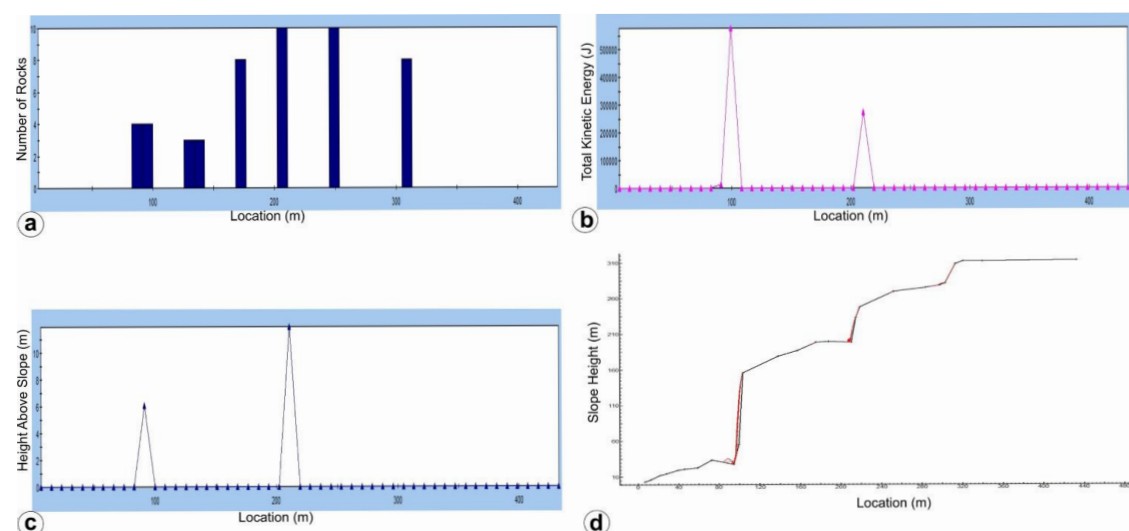


**Fig Fig.15** An example for the rockfall analyses results (a) runout distance (b) total kinetic energy (c) bounce height (d) the typical
rockfall trajectory





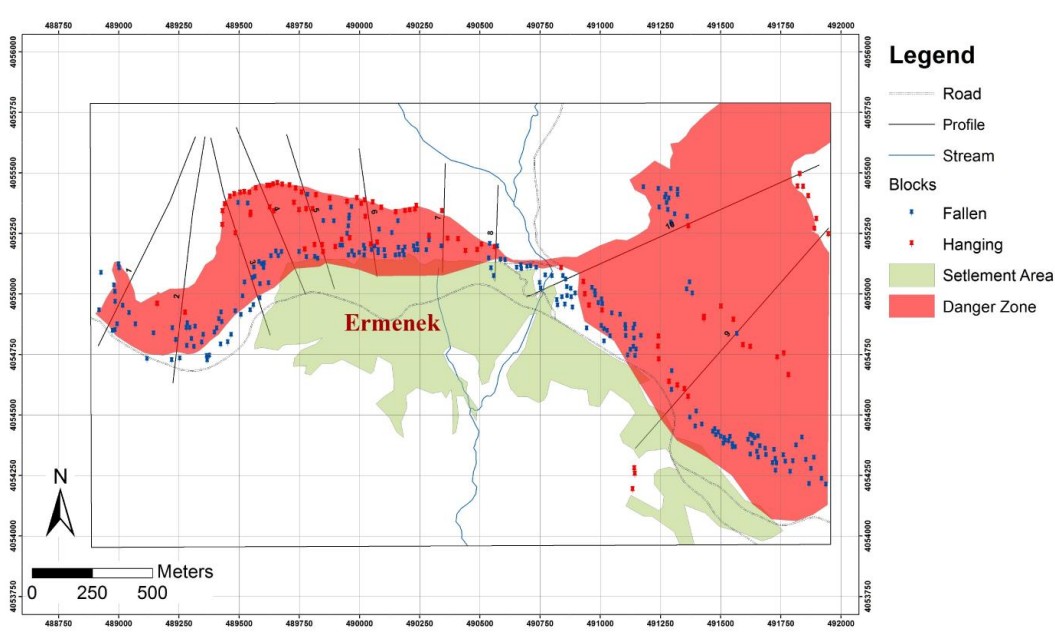

**Fig.16** The map showing the rockfall danger zone of study area