# Peer review of "Assessment of rockfall hazard on the steep-high slopes"

_Natural Hazards and Earth System Sciences, 2015_

## Referee Comment (RC1) · Anonymous Referee #1 · 12 Feb 2016

This paper presents a case study of rockfall analysis. The content of the paper should be considered as a technical report for a given case study not as a research paper. The authors did not succeeded in exposing novel modelling techniques, field investigations, or hazard zonation and management techniques. For that reason, I do not recommend publication of the paper despite it is a rather well presented case study done following rather state of the art techniques

Specific comments: 1) From line 146 to line 199, detailed field investigations are done by authors using "state of the art" techniques. These investigations may be interesting. However, in the context of the study, they are only used to determine the location of the potential release zone. I have the feeling that was not worth doing such detailed

investigations. The authors should absolutely clarify the aim of these investigations. 2) The rockfall analysis presented is very classical. The modelling approach used is based on 2D numerical classical models. The values of the different parameters of the model used are generally not well justified. In particular, the authors use restitution coefficients calibrated from back-analysis that is not detailed at all (what was the experimental or survey protocol? is it relevant for all materials, rock size, topographical configurations . . .?). 3) The results from the rockfall analysis are used to build a hazard map considering "maximum runout distance". The methodology for the building of the hazard map from the simulation results is not detailed and seems to be very rough (what does the maximum runout distance mean? why not combining rock energy with passing frequency? how do the authors used results along profiles to build a 3D map?) 4) The results from the analysis are also used to derive recommendations. The authors state that the hazard level is too high for protection measures to be efficient. This conclusion is not proven and not discussed.
* * *

---

## Author Comment (AC1) · 24 Feb 2016

REPLY OF THE AUTHORS TO ANONYMOUS REVIEWER'S COMMENTS

Reviewer Comment:

This paper presents a case study of rockfall analysis. The content of the paper should be considered as a technical report for a given case study not as a research paper. The authors did not succeeded in exposing novel modelling techniques, field investigations, or hazard zonation and management techniques.

Reply of the Authors:

Assessment of natural hazards such as rockfall, landslides, debris flow etc. are needed to be in a certain area. Therefore this study were carried out a serious rockfall prone area. If the researchers need to developed new techniques or parameters, rockfall studies to get out of a case study and working in a particular region. All these reason, we think that the manuscript is appropriate as a research paper.

In this manuscript we used existing rockfall techniques on a selected area. Determination and explanation of rockfall mechanism and affecting structural feature for rockfall events on the settlement area are main goal of this study. Developing novel modeling techniques, field investigation or hazard zonation and management techniques were not goal of this study. Furthermore, every research paper is not need to succeed to develop new techniques.

Reviewer Comment:

From line 146 to line 199, detailed field investigations are done by authors using "state of the art" techniques. These investigations may be interesting. However, in the context of the study, they are only used to determine the location of the potential release zone. I have the feeling that was not worth doing such detailed investigations. The authors should absolutely clarify the aim of these investigations.

Reply of the Authors:

Detailed field investigations are necessary for this study because the author's aim to understand of rockfall mechanism, relevant with lithological features. Moreover, all these parameters determining field study and laboratory tests are used rockfall analysis. From line 249 to line 252 rockfall mechanism in the study area was explain.

The aim of these investigations are clarified from line 16 to line 25.

Reviewer Comment:

The rockfall analysis presented is very classical. The modelling approach used is based on 2D numerical classical models. The values of the different parameters of

the model used are generally not well justified. In particular, the authors use restitution coefficients calibrated from back-analysis that is not detailed at all (what was the experimental or survey protocol? is it relevant for all materials, rock size, topographical configurations : : :?).

Reply of the Authors:

There are similar research papers that they have been refered many reserchers existing in literature using 2D numerical models to simulate and modeling rockfall events. (Hoek 1987; Pfeiffer and Bowen 1989; Azzoni et al. 1995; Jones et al. 2000; Guzetti et al. 2002, Guzetti et al, 2003; Agliardi and Crosta 2003; Schweigl et al 2003; Perret et al 2004; Yilmaz et al. 2008, Tunusluoglu and Zorlu 2009, Binal and Ercanoglu 2010; Zorlu et. al 2011; Katz et al 201175 ;Topal et al 2012; Chen et al. 1994; Keskin 2013, etc….). In this context, many researcers think that using 2D modelling for interprete rockfall events is not a classical model so, it is just a commonly prefered method. One of important parameter for rockfall anaysis are coefficient restitutions. These parameters were determined in the field via back analysis. While back analysis carried out topographical feature, size of rolling blocks, slope properties, were considered for each profile. The results of the coefficient restitutions value given a table (Table 5 added in the manuscript). Authors used to average value of the coefficient restitutions in rockfall analysis.

Reviewer Comment:

The results from the rockfall analysis are used to build a hazard map considering "maximum runout distance". The methodology for the building of the hazard map from the simulation results is not detailed and seems to be very rough (what does the maximum runout distance mean? why not combining rock energy with passing frequency? how do the authors used results along profiles to build a 3D map?)

Reply of the Authors:

The 2D rockfall danger zone map was produced by using ArcGIS 9.3 software ESRI 2009) from 10 profile. (the results obtained from rockfall analyses considering maximum runout distance of falling blocks.

ESRI (2009) ArcGIS version 9.3. 380 New York Street, Redlands, CA 92373-8100 USA.

Reviewer Comment:

The results from the analysis are also used to derive recommendations. The authors state that the hazard level is too high for protection measures to be efficient. This conclusion is not proven and not discussed.

Reply of the Authors:

Results and conclusion part of the manuscript revised as requested by the reviewer (from line 294 to line 297)

Please also note the supplement to this comment:
http://www.nat-hazards-earth-syst-sci-discuss.net/nhess-2015-337/nhess-2015-337-AC1-supplement.zip

---

## Referee Comment (RC2) · Anonymous Referee #2 · 4 Mar 2016

The paper presents a geotechnical investigation of a rockfall endangered area. While illustrating the hazard problem of the study case its content cannot be considered research. Albeit that the paper highlights an important hazard case that is worth investigating for the purposes of hazard risk reduction. The paper does not contain any new or novel methods to the scientific field of rockfall. The content is on the whole well presented with a structured report. However there are a number of questionable results presented in the rockfall analysis and the methodology and justifications for various assumptions therein are poorly discussed. Given these main points I do not recommend this manuscript for publication.

Specific comments: 1) The subject matter of rockfall hazards falls within the scope of

[Figure]

NHESS. However, the methodologies applied all be they current engineering practice, they do not represent the state of the art in rockfall science, and do not reveal anything new.

2) The methods applied are of standard geotechnical engineering practice and do not present any advances in technology, analysis techniques or methodology.

3) The authors provide a brief overview of rockfall models available to practicing geotechnical engineers. The review of rockfall models is by no means exhaustive and many of the current state of the art methods are missing from this review (e.g. Leine et al., 2013; Andrew et al., 2012; Basson, 2012 ). The model selected is a "rebound model" and assumes a spherical body for the rock applying restitution coefficients and friction to simulate ground contact. This method is full of uncertainty, in particular where factors such as rock shape and terrain roughness are forced into the contact parameters represented as restitution values (Lines 211 – 213). It is known that coefficients of restitution in rockfall modelling show a high degree of scatter and values above unity (see Buzzi et al., 2012 and Bourrier et al., 2012). Although there has been some attempt to provide a calibration of the coefficients of restitution applied in this work; the justification based on the field tests to determine the coefficients of restitution is vague and there is no discussion as to the selected range of restitution values that are applied randomly during the simulations. How are the contact parameters applied stochastically in the rockfall model? How does that selection of a spherical model affect the results? Would the large blocks in the images in figure 9 even become mobile unless they were spherical as is modelled?

4) The selection of block sizes is justified as they fall within the minimum and maximum measured values of detachable blocks. But how is this representative of the entire hazard case, what is the distribution of block sizes for the area? The applied methods to investigate the rock slope are extensive, including a discontinuity and thin section analysis. Indeed these are interesting; however it is not clear how these investigations relate to the main analyses performed in this paper. To a large extent these analyses

seem unnecessary. What is the significance of the joint spacing analysis, how are the joints controlling the block size? Are the blocks that are defined by jointing representative of the blocks that are impacting the buildings at the base of the slope? With the maximum energies given in table 4 it is likely that the rocks will fragment on impact with the slope. Can the given energies realistically be expected to impact the settlements at the base of slope? The application of the rockfall model requires that a 2-D profile is drawn over the terrain. What is the justification for the selected profiles? How does this deal with the likely lateral dispersion of rockfalls?

5) The results present the hazard case for the settlement below the slope, and are used to argue for the proposed mitigation solutions of evacuating the area. This may be justified, but it is difficult to reliably discern this from these analyses. The kinetic energies in table 4 are given in kJ, however the values given for maximum and minimum kinetic energy are far and beyond what is plausible for a rock in free fall from the given slope heights (see attached file). Can it be that the authors have given values in Joules as opposed to Kilo Joules as is stated in the header of the table? Based on this assumption and converting the given values to kilo Joules there are still discrepancies with the results. If the simulation results are compared to the maximum potential energy for the slope there are instances where the simulated kinetic energy exceeds the potential energy for the slope (Profile 6). Moreover, if the translational kinetic energy is computed from the given velocities they do not match the values given for maximum kinetic energy. This suggests that a component of rotational kinetic energy is contained in the maximum kinetic energy values quoted. The rotational aspect of rockfall is not discussed. How is the rotational component of rockfall computed in the model?

6) How is the conclusion that common rockfall mitigation methods are not suitable for this particular site justified? This is not clear in the text. Is the statement in line 269 -270 based on the output of the rockfall model or observations of rockfall?

7) The authors present in figure 13 an image of the back analysis applied to ascertain the coefficients of normal and tangential restitution; details of this methodology are

lacking. Commonly restitution coefficients are derived from the ratio of the rebound to the incident velocity of a given impact. It appears that video analysis has been applied to attain the velocities. However, firstly this is not clear nor are any details given as to how the images have been scaled to extract the velocity. What was the frame rate of the videos used in these measurements, what was the depth of field of the passing block? Such factors have a strong bearing on the velocity that is extracted, respectively the resultant restitution coefficients. How many experiments were performed to attain the distributions of the restitution coefficients given? Restitution coefficients measured by velocity assuming a point mass are renowned for large scatter, which is an effect of the rock's shape.

11) On the whole the symbols and abbreviations are correct. Table 4 contains kinetic energy values labelled as kJ that far exceeds that which is likely for the described rockfall case. It is assumed that the authors are quoting their results in Joules as opposed to kilo Joules.

References: Andrew, R., Hume, H., Bartingale, R., Rock, A., and Zhang, R. (2012). CRSP-3D Users Manual Colorado Rockfall Simulation Program.

Basson, F. R. P. (2012). Rigid body dynamics for rock fall trajectory simulation. American Rock Mechanics Association, 46th US Rock Mechanics / Geomechanics Symposium, Chicago, IL, USA, pages 1-7.

Bourrier, F., Berger, F., Tardif, P., Dorren, L., and Hungr, O. (2012). Rockfall rebound: comparison of detailed _eld experiments and alternative modelling approaches. Earth Surface Processes and Landforms, 37(6):656-665.

Buzzi, O., Giacomini, A., and Spadari, M. (2012). Laboratory investigation on high values of restitution coefficients. Rock Mechanics and Rock Engineering, 45:35-43.

Leine, R., Schweizer, A., Christen, M., Glover, J., Bartelt, P., and Gerber, W. (2014). Simulation of rockfall trajectories with consideration of rock shape. Multibody System

Dynamics, 32:241-271.

Please also note the supplement to this comment:
http://www.nat-hazards-earth-syst-sci-discuss.net/nhess-2015-337/nhess-2015-337-RC2-supplement.pdf
* * *
[Figure]

**Supplement:**

| Profile | Potential Height | mass block kg | Max kE (given in kJ) | kJ assuming that joules are given | PE kJ | Difference of PE to given kJ | Difference of PE to kJ (Assuming kJ is given as Joules) | Max Velocity | Maximum free fall velocity (m/s) | kE of Velocity | Maximum kE of freefall velocity | Difference between the given kE (Assuming Jouels mistake) to the kE of the given velocity | Difference between given velocity and free fall velocity (m/s) |
|---|---|---|---|---|---|---|---|---|---|---|---|---|---|
| 1 | 88 | 100 | 14000 | 14 | 86.328 | -13913.672 | 72.328 | 16 | 41.55 | 12.8 | 86.328 | -1.2 | 25.55 |
| | 88 | 1000 | 120000 | 120 | 863.28 | -119136.72 | 743.28 | 14 | 41.55 | 98 | 863.28 | -22 | 27.55 |
| | 88 | 10000 | 1400000 | 1400 | 8632.8 | -1391367.2 | 7232.8 | 16 | 41.55 | 1280 | 8632.8 | -120 | 25.55 |
| 2 | 33 | 100 | 2200 | 2.2 | 32.373 | -2167.627 | 30.173 | 6.3 | 25.45 | 1.9845 | 32.373 | -0.2155 | 19.15 |
| | 33 | 1000 | 18200 | 18.2 | 323.73 | -17876.27 | 305.53 | 7.21 | 25.45 | 25.99205 | 323.73 | 7.79205 | 18.24 |
| | 33 | 10000 | 62033 | 62.033 | 3237.3 | -58795.7 | 3175.267 | 3.27 | 25.45 | 53.4645 | 3237.3 | -8.5685 | 22.18 |
| 3 | 334 | 100 | 14027 | 14.027 | 327.654 | -13699.346 | 313.627 | 16.29 | 80.95 | 13.268205 | 327.654 | -0.758795 | 64.66 |
| | 334 | 1000 | 1750000 | 1750 | 3276.54 | -1746723.46 | 1526.54 | 46.3 | 80.95 | 1071.845 | 3276.54 | -678.155 | 34.65 |
| | 334 | 10000 | 11200000 | 11200 | 32765.4 | -11167234.6 | 21565.4 | 63.5 | 80.95 | 20161.25 | 32765.4 | 8961.25 | 17.45 |
| 4 | 145 | 100 | 34500 | 34.5 | 142.245 | -34357.755 | 107.745 | 28.43 | 53.34 | 40.413245 | 142.245 | 5.913245 | 24.91 |
| | 145 | 1000 | 583400 | 583.4 | 1422.45 | -581977.55 | 839.05 | 34.42 | 53.34 | 592.3682 | 1422.45 | 8.9682 | 18.92 |
| | 145 | 10000 | 6973000 | 6973 | 14224.5 | -6958775.5 | 7251.5 | 33.25 | 53.34 | 5527.8125 | 14224.5 | -1445.1875 | 20.09 |
| 5 | 123 | 100 | 64300 | 64.3 | 120.663 | -64179.337 | 56.363 | 36.32 | 49.12 | 65.95712 | 120.663 | 1.65712 | 12.80 |
| | 123 | 1000 | 670000 | 670 | 1206.63 | -668793.37 | 536.63 | 33.24 | 49.12 | 552.4488 | 1206.63 | -117.5512 | 15.88 |
| | 123 | 10000 | 6270000 | 6270 | 12066.3 | -6257933.7 | 5796.3 | 33.5 | 49.12 | 5611.25 | 12066.3 | -658.75 | 15.62 |
| 6 | 103 | 100 | 102500 | 102.5 | 101.043 | -102398.957 | -1.457 | 46 | 44.95 | 105.8 | 101.043 | 3.3 | -1.05 |
| | 103 | 1000 | 958000 | 958 | 1010.43 | -956989.57 | 52.43 | 45 | 44.95 | 1012.5 | 1010.43 | 54.5 | -0.05 |
| | 103 | 10000 | 10500000 | 10500 | 10104.3 | -10489895.7 | -395.7 | 43.5 | 44.95 | 9461.25 | 10104.3 | -1038.75 | 1.45 |
| 7 | 336 | 100 | 8320 | 8.32 | 329.616 | -7990.384 | 321.296 | 12 | 81.19 | 7.2 | 329.616 | -1.12 | 69.19 |
| | 336 | 1000 | 83000 | 83 | 3296.16 | -79703.84 | 3213.16 | 11.5 | 81.19 | 66.125 | 3296.16 | -16.875 | 69.69 |
| | 336 | 10000 | 7000000 | 7000 | 32961.6 | -6967038.4 | 25961.6 | 11.5 | 81.19 | 661.25 | 32961.6 | -6338.75 | 69.69 |
| 8 | 104 | 100 | 14300 | 14.3 | 102.024 | -14197.976 | 87.724 | 16.7 | 45.17 | 13.9445 | 102.024 | -0.3555 | 28.47 |
| | 104 | 1000 | 870000 | 870 | 1020.24 | -868979.76 | 150.24 | 45 | 45.17 | 1012.5 | 1020.24 | 142.5 | 0.17 |
| | 104 | 10000 | 8650000 | 8650 | 10202.4 | -8639797.6 | 1552.4 | 42 | 45.17 | 8820 | 10202.4 | 170 | 3.17 |
| 9 | 95 | 100 | 72000 | 72 | 93.195 | -71906.805 | 21.195 | 38 | 43.17 | 72.2 | 93.195 | 0.2 | 5.17 |
| | 95 | 1000 | 630000 | 630 | 931.95 | -629068.05 | 301.95 | 36 | 43.17 | 648 | 931.95 | 18 | 7.17 |
| | 95 | 10000 | 7120000 | 7120 | 9319.5 | -7110680.5 | 2199.5 | 37 | 43.17 | 6845 | 9319.5 | -275 | 6.17 |
| 10 | 68 | 100 | 24500 | 24.5 | 66.708 | -24433.292 | 42.208 | | 36.53 | 0 | 66.708 | -24.5 | 36.53 |
| | 68 | 1000 | 653000 | 653 | 667.08 | -652332.92 | 14.08 | 34.5 | 36.53 | 595.125 | 667.08 | -57.875 | 2.03 |
| | 68 | 10000 | 6250000 | 6250 | 6670.8 | -6243329.2 | 420.8 | 34 | 36.53 | 5780 | 6670.8 | -470 | 2.53 |